# MSTI-Plus: Introducing Non-Sarcasm Reference Materials to Enhance Multimodal Sarcasm Target Identification

## ABSTRACT

Sarcasm is a subtle expression that indicates the incongruity between literal meanings and factual opinions. For multimodal posts in social medias which consist of both images and texts, sarcasm expressions are even more widespread. Recent works have paid attentions to Multimodal Sarcasm Target Identification (MSTI), which focuses on detecting aspect terms of mockery or ridicule as sarcasm targets. However, the current MSTI benchmark only contains annotations on fine-grained sarcasm targets within sarcastic samples. In practice, it will be featured by two major limitations. First, there lack annotations on non-sarcasm aspects to inform deep models to perceive the semantic difference between sarcasm targets and non-sarcasm aspects. As a result, deep models will tend to incorrectly recognize non-sarcasm aspects as sarcasm targets. Second, there lack non-sarcasm samples to inform deep models to perceive the inherent semantics of sarcasm intentions. Due to the subtle characteristic of sarcasm expressions, models trained with only fine-grained supervision signals cannot thoroughly understand the sarcasm semantics, making the fine-grained task of sarcasm target identification restricted. Motivated by these limitations, this work reconstructs a more comprehensive MSTI benchmark by introducing both fine-grained non-sarcasm aspect annotations for existing sarcasm samples and non-sarcastic samples as non-sarcasm references to enable deep models to clearly perceive the mentioned information during training. Based on the multi-granularity (i.e., both aspect-level and sample-level) non-sarcasm information introduced into this new benchmark, this work further proposes a pluggable Semantics-aware Sarcasm Target Identification mechanism to enhance sarcasm target identification by modeling the overall semantics of sarcasm intentions via an auxiliary sample-level sarcasm recognition task. By modeling the overall semantics of sarcasm intention, deep models can obtain a more comprehensive understanding on sarcasm semantics, leading to improved performance on fine-grained sarcasm target identification. Extensive experiments are conducted to validate our contribution. Both the dataset and implementation code will be released once the paper is accepted.

**Relevance Statement:** This work aims to provide a solid foundation for user sentiment analysis on social medias by reducing the interference of subtle sentiment expressions which are widely widespread in webs.

## CCS CONCEPTS

• **Information systems** → **Sentiment analysis**; **Multimedia information systems**.

## KEYWORDS

Multimodal sarcasm target identification, social media analysis, sentiment analysis, multimodal deep learning.

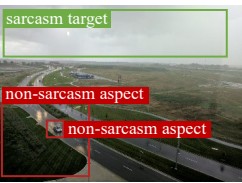 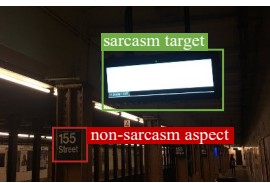

perfect weather for the eclipse today here in kc. #eclipse2017

<user> oh good! i was wondering when the next train was arriving! you're always so helpful …, mta.

**Figure 1: Examples for sarcasm samples containing both sarcasm aspects (shown in the green color) and non-sarcasm aspects (shown in the red color). Left: the cloudy weather within the image is contrary to the textual description "perfect weather". Right: the negative information conveyed by "blank train arrival schedule" within the image is contrary to the positive sentiment conveyed by "the helpful work of the transportation organization dubbed mta" within the text.**

## 1 INTRODUCTION

Sarcasm is a subtle form of sentiment expression where the literal meanings contradict the factual opinions of people [9]. As the sarcastic utterances frequently appear on social media platforms, sarcasm detection receives considerable attentions and plays a crucial role in various social media analysis applications such as sentiment analysis [24] or public opinion mining [26]. With the rapid development of social platforms, users tend to share multimodal posts consisting of images and texts onto social medias like Twitter or Facebook. Under this background, researchers begin to focus on multimodal sarcasm detection [2, 4, 19, 21, 27, 37, 39], which leverages both visual and textual modalities to determine whether a post conveys the sarcastic sentiment. Compared with textual sarcasm detection, multimodal sarcasm detection models can further leverage the incongruity information between image and text to mine the sarcasm intention and hence achieve enhanced performance. Recently, researchers further pose the fine-grained sarcasm detection task dubbed Multimodal Sarcasm Target Identification (MSTI) [36], aiming at detecting aspect terms of mockery or ridicule as sarcasm targets within sarcastic multimodal samples.

Sarcasm target identification is important for understanding sarcasms in depth, as well as improving the interpretability for sarcasm detection. Existing works implement multimodal sarcasm target identification mainly based on the MSTI benchmark released in [36]. The MSTI benchmark consists of multimodal sarcasm samples with fine-grained annotations for both visual and textual sarcasm targets. Based on the released MSTI benchmark, existing works train multimodal deep models to identify sarcasm targets within sarcastic samples. However, as shown in Figure 1, there usually exist non-sarcasm aspects (shown in the red color) that do not convey the sarcasm intention in sarcastic samples. As the current MSTI benchmark does not contain the supervision signal of non-sarcasm aspects, the trained models cannot explicitly perceive the semantic

difference between sarcasm targets and non-sarcasm aspects. As a result, the trained models may incorrectly treat sarcasm target identification as a common aspect term extraction task [17, 18, 23] and tend to incorrectly recognize non-sarcasm aspects as sarcasm targets (as will be shown in Figure 6 of our experiments). Motivated by the above observation, this work takes a further step to include fine-grained annotations of non-sarcasm aspects into the benchmark. To this end, we manually annotate the non-sarcasm aspects for samples of the current MSTI benchmark. Supervised by the fine-grained information of both sarcasm targets and non-sarcasm aspects, deep models can explicitly perceive the semantic difference between sarcasm targets and non-sarcasm aspects, leading to clearly improved performance for sarcasm target identification.

Moreover, in practice, sarcasm is a comprehensive sentiment expression which should be understood by considering the overall semantics of samples. Only fine-grained supervision within sarcasm samples cannot effectively guide deep models to thoroughly understand the sarcasm semantics, which in turn restricts deep models' ability in the fine-grained sarcasm target identification task. Hence, we consider integrating the sample-level supervision of sarcasms as a higher-level guidance to lead deep models to better understand the inherent semantics of sarcasm intentions. With this consideration, our work further introduces non-sarcastic samples as the sample-level non-sarcasm references. To this end, non-sarcastic multimodal samples with fine-grained annotations on non-sarcasm aspects re-organized from the existing Grounded Multimodal Named Entity Recognition (GMNER) benchmark [41] are incorporated into the current MSTI benchmark. We have manually checked that all the incorporated samples do not convey the sarcasm intention. With both extra fine-grained annotations on non-sarcasm aspects of existing sarcasm samples and the newly incorporated non-sarcasm samples, we coin the reconstructed benchmark as MSTI-Plus.

Based on the multi-granularity (i.e., both aspect-level and sample-level) non-sarcasm information introduced in the reconstructed MSTI-Plus benchmark, we further propose a pluggable Semantics-aware Sarcasm Target Identification (SaSTI) mechanism, which can be flexibly attached on top of existing multimodal sarcasm target identification models. As motivated by the above discussion, the core idea of the proposed SaSTI mechanism mainly focuses on implementing fine-grained sarcasm target identification under the guidance of the overall understanding for sarcasm expressions. To this end, a sample-level sarcasm identification task is introduced on top of sample features to inform the overall understanding for sarcastic expressions. Specifically, to model the overall semantics of sarcasm intentions, a semantic memory is dynamically maintained during training by performing moving average on sample-level features of sarcasm expressions. Afterwards, the semantic memory will be utilized to inform specific sarcasm targets of textual tokens or visual objects, making the fine-grained sarcasm target identification performed with the guidance of the overall understanding for sarcasm intentions. By introducing both the new benchmark and new method, this work has the following advantages compared to existing works [20, 36]. First, with fine-grained supervision signals of both sarcasm targets and non-sarcasm aspects, deep models can explicitly perceive the semantic difference between them, preventing from incorrectly treating sarcasm target identification as a common aspect term extraction task. Second, by modeling the

overall semantics of sarcasm intentions with the aid of sample-level non-sarcasm references, deep models can obtain a more comprehensive understanding for sarcasm expressions, leading to improved performance on the fine-grained target identification task. Extensive experiments have been conducted to validate the contribution of this work.

To sum up, the main contributions of this work are listed as follows:

- This work draws the first attention on the limitation of the current MSTI benchmark, including: 1) lacking annotations on non-sarcasm aspects to inform deep models to perceive the semantic difference between sarcasm targets and non-sarcasm aspects; 2) lacking non-sarcasm samples to inform deep models to perceive the inherent semantics of sarcasm intentions.
- This work proposes a more comprehensive benchmark by introducing both fine-grained non-sarcasm aspect annotations for existing sarcastic samples and non-sarcastic samples, which enables deep models to more clearly perceive the inherent semantics of sarcasms with the aid of supervision signals provided by the introduced non-sarcasm references.
- Based on the multi-granularity non-sarcasm references introduced in our reconstructed benchmark, this work further proposes the pluggable SaSTI mechanism to enhance multimodal sarcasm target identification based on the guidance of the overall understanding for sarcasm intentions.
- Extensive experiments are conducted based on our proposed benchmark. The experimental results clearly demonstrate the advantages brought by this work.

## 2 RELATED WORKS

**Sarcasm Detection.** Sarcasm detection leverages the incongruity of sentiment within contexts to mine sarcastic intentions. Initially, researchers primarily focus on the text modality, applying a variety of techniques ranging from feature engineering to deep neural networks to detect incongruity information in texts [8, 10, 16, 33, 34, 38, 42]. For example, Tay et al. [34] and Xiong et al. [38] model incongruous interactions between individual words for sarcasm detection by using attention-based neural networks. Babanejad et al. [1] conduct sarcasm detection by extending the architecture of BERT to mine sarcastic intentions. With the rapid development of social platforms, multimodal posts consisting of text and images are widely shared on social medias. Under this background, multimodal sarcasm detection has received increasing attentions and a series of valuable works have emerged [14, 27, 30, 35, 37, 39]. In particular, Liang et al. [19] introduce the cross-modal graph to shape the sarcastic relations across the image and text modalities. Wen et al. [37] propose a dual perceiving architecture to model the incongruity between texts and images from the factual and sentiment views. Qin et al. [30] leverage CLIP [31] to mine sarcasm cues from the text, image, and text-image interaction views.

**Sarcasm Target Identification.** To further understand sarcasms in depth, researchers have recently introduced the task of fine-grained sarcasm detection. Early works mainly focus on detecting sarcasm targets in texts [15, 28, 29]. With the growing number

of multimodal posts on social media platforms, models that rely solely on the text modality face challenges in detecting sarcastic targets within multimodal posts. To this end, researchers begin to explore fine-grained multimodal sarcasm detection [7, 36]. In particular, Wang et al. [36] propose the Multimodal Sarcasm Target Identification (MSTI) task and release a benchmark consisting of multimodal sarcasm samples with fine-grained annotations on both textual and visual sarcasm targets. Their proposed approach utilizes a cross-modal attention mechanism to detect sarcasm targets within texts and images. Lin et al. [20] further propose to enhance sarcasm target identification by generating explanations for sarcasms as contextual information.

However, existing works on multimodal sarcasm target identification are primarily based on the MSTI benchmark released in [36], which only involves annotations on sarcasm targets of sarcastic samples. Due to the lack of fine-grained annotations on non-sarcasm aspects, it is hard to perceive the semantic difference between sarcasm targets and non-sarcasm aspects. As a result, models based on the MSTI benchmark may incorrectly treat sarcasm target identification as a common aspect term extraction task and tend to incorrectly recognize normal non-sarcasm aspects as sarcasm targets. On the other hand, only fine-grained supervision within sarcasm samples cannot inform deep models to thoroughly perceive the inherent semantics of sarcasm intentions, making the fine-grained task of sarcasm target identification restricted. Motivated by this limitation, this work constructs the MSTI-Plus benchmark by further introducing both aspect-level and sample-level non-sarcasm references into the dataset. With the newly introduced annotations on non-sarcasm aspects, deep models trained on the MSTI-Plus benchmark can more explicitly perceive the semantic difference between sarcasm targets and non-sarcasm aspects. Moreover, with the introduced non-sarcasm samples as sample-level non-sarcasm references, deep models can be trained to perceive the overall semantics of sarcasm intentions, which can be utilized to provide positive supports for fine-grained sarcasm target identification.

## 3 THE MSTI-PLUS BENCHMARK

In order to enable deep models to focus on perceiving the semantic difference between sarcasm targets and normal non-sarcasm aspects, this work introduces a more comprehensive multimodal sarcasm target identification benchmark dubbed MSTI-Plus, which involves fine-grained annotations on both sarcasm targets and normal non-sarcasm aspects. In general, multimodal sarcasm target identification mainly involves two major subtasks, i.e., textual sarcasm target identification and visual sarcasm target identification. For the visual sarcasm target identification subtask, the current MSTI benchmark treats it as an object detection task, which focuses on detecting the bounding boxes of sarcasm targets from images. However, based on our empirical experiments, we find that end-to-end object detectors are usually hard to train for this identification task which involves subtle and complex human sentiments. Moreover, the main focus for visual sarcasm target identification lies in detecting visual sarcasm targets to provide interpretabilities for existing sarcasm detection systems [19, 21, 35], rather than accurately detecting their bounding boxes as a precision-demanding visual task such as instance segmentation or object detection in automatic drive. Hence, this work advocates to perform the visual sarcasm

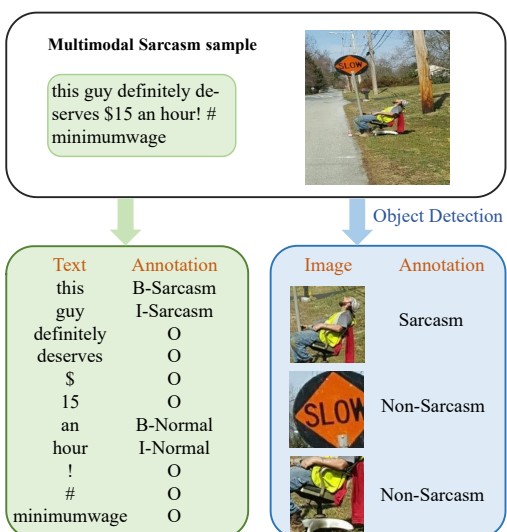

**Figure 2: Example for multimodal data with fine-grained annotations on both sarcasm targets and non-sarcasm aspects.**

target identification subtask as a classification problem based on visual targets extracted from external object detectors, i.e., identify whether a visual target conveys the sarcasm intention. Details for the MSTI-Plus dataset construction are as follows.

### 3.1 Data Collection

We collect available multimodal posts from the MSTI dataset [36] and the MNER dataset [41] to construct the MSTI-Plus dataset. Specifically, we collect 2,500 sarcastic image-text pairs from the MSTI dataset, which involve fine-grained labels for both textual and visual sarcasm targets annotated by Wang et al. [36]. In order to balance the number of different types of samples, we also collect 2,500 non-sarcastic multimodal posts from the MNER dataset as sample-level non-sarcasm references. For the 5,000 multimodal samples collected in our dataset, we further annotate fine-grained non-sarcastic aspects for both text and image modalities as aspect-level non-sarcasm references. For the image modality, we first adopt VinVL [43] which is a commonly-used object detection model to extract visual targets from images, and then annotate whether they are sarcasm targets. In this work, our annotators focus on manually checking the existing labels and annotating fine-grained non-sarcasm aspects for both texts and images.

### 3.2 Fine-grained Annotation

Our annotations focus on whether the textual aspects and visual objects of multimodal samples express the sarcastic intention or not. To this end, each textual and visual target is annotated with either a sarcastic or non-sarcastic label. As shown in Figure 2, we can see a lazy man within the image lying on the chair. This sample mainly conveys the negative sentiment for the man by using the sarcastic utterance. Hence, we annotate the phrase "this guy" as the sarcasm target according to the BIO (Beginning, Inside, Outside) regulation [32]. On the other hand, the phrase "an hour" that does not convey the sarcastic sentiment is annotated as a non-sarcasm aspect. For the image modality, the first visual region shown in the

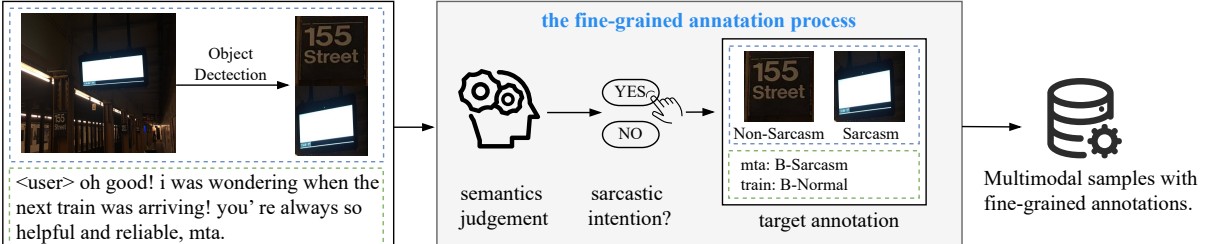

**Figure 3: The annotation process in which annotators perform the fine-grained annotation for a multimodal sample post. First, the raw text and image, as well as visual targets detected by external object detectors, are allocated to annotators. Second, the annotators check whether a sample conveys the sarcasm intention based on its semantic content. Afterwards, the annotators will label textual aspects and visual targets as sarcasm targets or normal non-sarcasm aspects according to their understanding.**

blue box will also be annotated as a sarcasm target. In contrast, the remaining visual regions will be annotated as non-sarcasm aspects.

Formally, the labels used to annotate targets in texts and images are as follows: 1) **B-Sarcasm:** indicates the beginning of a sarcasm target consisting of a word or a phrase; 2) **I-Sarcasm:** denotes an inside part of a sarcasm target consisting of a phrase; 3) **B-Normal:** indicates the beginning of a normal non-sarcasm aspect, representing that the word does not convey the sarcastic intention; 4) **I-Normal:** denotes an inside part of a normal non-sarcasm aspect consisting of a phrase; 5) **O:** indicates that the word does not belong to an aspect term; 6) **Sarcasm:** indicates that a detected visual target conveys the sarcastic intention; 7) **Non-Sarcasm:** denotes that an extracted visual target does not carry the sarcastic meaning. Among the above labels, B-Sarcasm, I-Sarcasm, B-Normal, I-Normal, and O are used to annotate the textual modality, while Sarcasm and Non-Sarcasm are used to annotate the image modality.

## 3.3 Annotation Process

Given a multimodal post, the participated annotators apply the corresponding labels mentioned above to annotate the textual and visual modality, respectively. The annotation process is shown in Figure 3. The annotators first check whether a sample post conveys the sarcasm intention based on its semantic information. Afterwards, the annotators label textual aspects and visual targets as sarcasm targets or normal non-sarcasm aspects. To ensure the annotation quality, each multimodal post is labeled by three annotators. In the annotation process, we face two major challenges, including 1) **the limited contents of sample posts:** solely depending on the sample content, the annotators have limitations in accurately understanding the sarcastic intention without extra background knowledge; 2) **the annotation discrepancy due to the subjective judgement for sarcasm contents:** as the sarcasm targets usually subtly exist in multimodal posts, the annotations will show understanding discrepancies across annotators. To address the first problem, each annotator will explore the background contents corresponding to the sample to be annotated, which enables the annotators to conduct a more reasonable annotation. For the second challenge, we establish a two-round annotation agreement to minimize the subjectivity of annotators. Specifically, in the first-round, if at least two annotators agree the annotation for a textual word or visual target, the corresponding fine-grained annotation will be accepted. Samples having rejected fine-grained annotations will be re-labeled by other three annotators via a second-round annotation

**Table 1: The statistics of the MSTI-Plus benchmark.**

| Split | #Tweet | #Textual aspect | | #Visual target | |
|-------|--------|---------|-------------|---------|-------------|
| | | Sarcasm | Non-sarcasm | Sarcasm | Non-sarcasm |
| Train | 3,062 | 1,490 | 4,158 | 897 | 7,243 |
| Dev | 612 | 285 | 854 | 165 | 1,426 |
| Test | 614 | 297 | 830 | 225 | 1,433 |
| Total | 4,288 | 2,072 | 5,842 | 1,287 | 10,102 |

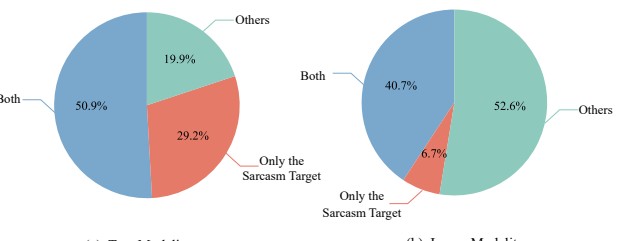

(a) Text Modality          (b) Image Modality

**Figure 4: Proportions of different sarcastic sample types based on the presence of sarcasm targets and normal non-sarcasm aspects. The notation "Both" indicates the proportion of sarcastic samples containing both sarcasm targets and normal non-sarcasm aspects within the corresponding modality, "Only the Sarcasm Target" indicates the proportion of sarcastic samples containing only sarcasm targets within the corresponding modality, and "Others" indicates the proportion of sarcastic samples containing no sarcasm targets within the corresponding modality.**

agreement process. Only the sample that passes the above annotation agreement process can be placed into our dataset, otherwise it will be discarded. We perform the quality control work to ensure the effectiveness of data (shown in Section A of supplementary).

## 3.4 Dataset Analysis

In Table 1, we show the statistics of our dataset. After the above annotation agreement process, 4,288 samples are remained in our dataset. In this work, 3,062/612/614 tweets are respectively utilized as Train/Dev/Test samples in the experiments. Table 1 also shows the statistics for fine-grained text aspects and visual targets. It can be observed that both the text and image modalities contain a large amount of non-sarcasm aspects. Moreover, as shown in Figure 4, we also respectively visualize the proportions of different sarcastic sample types based on the presence of sarcasm targets and normal non-sarcasm aspects. Taking the text modality (shown in Figure 4 (a)) as an example, there include three cases: sarcastic

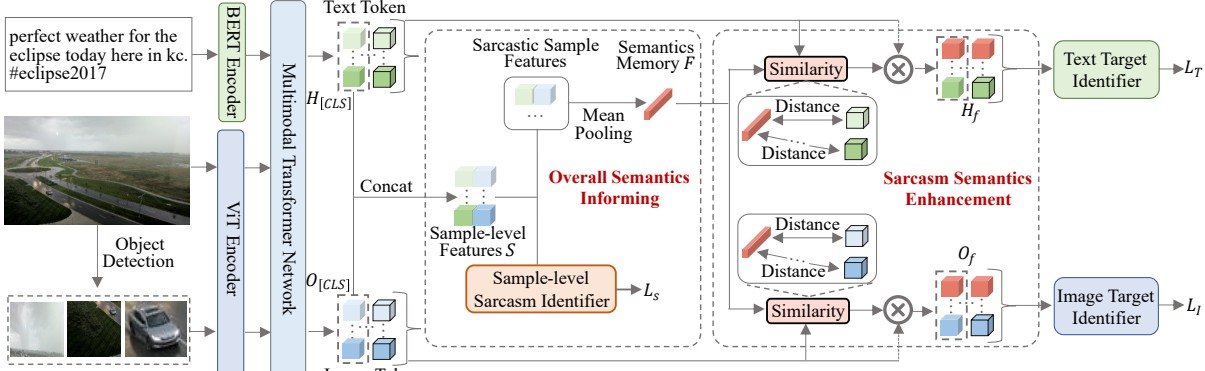

**Figure 5: Our proposed SaSTI mechanism attached on top of deep models. Specifically, our approach introduces a sample-level sarcasm identification task on top of sample features to inform comprehensive semantics of sarcasm expressions. Besides, a semantic memory is introduced to inform the textual token or the visual token with close distance to it. Afterwards, the semantic memory will be utilized to inform specific sarcasm targets of textual tokens or visual objects, which enables the fine-grained sarcasm target identification to perform with the guidance of the overall understanding for sarcasm intentions.**

samples containing both sarcasm targets and normal non-sarcasm aspects within the text modality, sarcastic samples containing only sarcasm targets within the text modality, and sarcastic samples containing no sarcasm targets within the text modality. The image modality (shown in Figure 4 (b)) is featured by similar cases. We can see that sarcasm targets and non-sarcasm aspects coexist within a large number of sarcastic samples, which shows the necessity of exploring the semantic difference across them. Hence, based on the above analysis, compared to solely leveraging sarcasm targets for training, models that consider non-sarcasm aspects during the training stage can be explicitly informed to perceive the semantic difference between sarcasm targets and non-sarcasm aspects, which can prevent from incorrectly recognizing non-sarcasm aspects as sarcasm targets.

## 4 SARCASM TARGET IDENTIFICATION WITH NON-SARCASM REFERENCES

### 4.1 Problem Statement

This work focuses on multimodal sarcasm target identification involving both sarcastic and non-sarcastic samples. Specifically, each sample contains a textual description $W_i$, an image $I_i$, and visual targets $P_i = \{p_{i,1}, p_{i,2}, \cdots, p_{i,j}\}$ ($j$ denotes the number of visual targets within a sample) extracted by an external object detecton model. The main purpose of multimodal sarcasm target identification is to learn an identification model $\mathcal{F}(W_i, I_i, P_i)$ by leveraging the fine-grained supervision information about sarcasm targets. After training, the identification model $\mathcal{F}(W_i, I_i, P_i)$ is expected to recognize fine-grained sarcastic labels for multimodal samples, i.e., $Y^T_{i,j} \in$ {B-Sarcasm, I-Sarcasm, B-Normal, I-Normal, O} for textual words and $Y^I_{i,j} \in \{0, 1\}$ for visual targets, where 1 represents that $p_{i,j}$ is a sarcastic target and vice versa.

### 4.2 Model Overview

Based on the multi-granularity (i.e., both aspect-level and sample-level) non-sarcasm references introduced in the reconstructed MSTI-Plus benchmark, this work further proposes an effective multi-task

framework which involves sarcastic supervision information of different levels to fully utilize the non-sarcasm reference materials. Figure 5 depicts the overall architecture of our training framework. First, we introduce fine-grained supervision of non-sarcasm aspects to train deep models, which enables deep models to explicitly perceive the semantic difference between sarcasm targets and normal non-sarcasm aspects. On the other hand, we introduce sample-level supervision of sarcasms as a higher-level guidance to encourage deep models to perceive the overall semantics of sarcasm expressions, which is then utilized to enhance the fine-grained multimodal sarcasm target identification task. To this end, we design a pluggable Semantics-aware Sarcasm Target Identification (SaSTI) mechanism, which can be flexibly appended on top of existing multimodal sarcasm target identification models (i.e., the "Multimodal Transformer Network" in Figure 5). Specifically, a sample-level sarcasm identification task is introduced on top of sample features to inform the overall understanding of sarcasms for MSTI enhancement. To model the overall semantics of sarcasm intentions, a semantic memory is dynamically maintained during training by performing moving average on sample-level features of sarcastic sample posts. Afterwards, the modeled semantic memory will be utilized to inform specific sarcasm targets respectively within the text and image modality, which enables the fine-grained sarcasm target identification task implemented based on the overall understanding for sarcasm semantics.

### 4.3 Multimodal Sample Processing

This work utilizes BERT [5] to process the textual description $P_i$ into textual features $\mathbf{M} = [\mathbf{m}_{[CLS]}, \mathbf{m}_1, \cdots, \mathbf{m}_n] \in \mathbb{R}^{(n+1)\times d}$, where $n$ and $d$ respectively represents the number of word tokens and the feature dimension. For the image modality, we utilize Vision Transformer (ViT) [6] to extract features respectively from each visual target, and then concatenate their [CLS] tokens as $\mathbf{R} = [\mathbf{r}^1_{[CLS]}, \mathbf{r}^2_{[CLS]}, \cdots, \mathbf{r}^j_{[CLS]}] \in \mathbb{R}^{j\times d}$, where $j$ represents the number of visual targets within $P_i$. Moreover, in order to provide the image context for these fine-grained visual targets, we also

utilize ViT to extract visual features from the whole image $I_i$, resulting in: $\mathbf{V} = [\mathbf{v}_{[CLS]}, \mathbf{v}_1, \mathbf{v}_2, \cdots, \mathbf{v}_a] \in \mathbb{R}^{(a+1) \times d}$, where $a$ represents the number of image tokens. Afterwards, visual features of both the whole image $\mathbf{V}$ and the fine-grained visual targets $\mathbf{R}$ will be concatenated to establish the final visual modality feature, which is denoted as $\mathbf{N} = [\mathbf{V}, \mathbf{R}] \in \mathbb{R}^{(a+j+1) \times d}$.

Afterwards, for each multimodal sample, its textual modality feature $\mathbf{M}$ and visual modality feature $N$ will be input into a followed multimodal transformer network to undergo a thorough cross-modal interactive process, resulting in mutually reinforced textual features $\mathbf{H} \in \mathbb{R}^{(n+1) \times d}$ and visual features $\mathbf{O} \in \mathbb{R}^{(a+j+1) \times d}$. The mentioned multimodal transformer network can be implemented flexibly. In our work, we utilize the existing architecture of either UMT [40] or MSTI [36] to implement it.

## 4.4 Semantics-aware Sarcasm Target Identification

The main idea of the SaSTI mechanism focuses on enhancing fine-grained sarcasm target identification based on the overall semantics of sarcasms informed by the introduced sample-level non-sarcasm references.

**Modeling Overall Semantics of Sarcasms.** Given the textual feature $\mathbf{H} \in \mathbb{R}^{(n+1) \times d}$ and visual feature $\mathbf{O} \in \mathbb{R}^{(a+j+1) \times d}$ output by the multimodal transformer network mentioned above, SaSTI introduces a sample-level sarcasm identification task on top of them to inform the overall semantics of sarcastic expressions. Specifically, the [CLS] tokens of $\mathbf{H}$ and $\mathbf{O}$ (i.e., $\mathbf{H}_{[CLS]}$ and $\mathbf{O}_{[CLS]}$ respectively corresponds to $\mathbf{m}_{[CLS]}$ within $M$ and $\mathbf{v}_{[CLS]}$ within $N$) are first concatenated to obtain sample-level multimodal features: $S \in \mathbb{R}^{1 \times 2d}$. Afterwards, $S$ will be fed into a Sample-level Sarcasm Identifier for predicting sample-level sarcastic labels. Supervised by sample-level information of sarcasms, $S$ will be trained to model the overall semantics of sarcasms:

$$\hat{y}_s = \text{Sigmoid}(\text{MLP}(S)),$$
$$\mathcal{L}_s = -[y_s \log(\hat{y}_s) + (1 - _s) \log(1 - \hat{y}_s)], \quad (1)$$

where MLP consists of one linear layer, $\hat{y}_s$ represents the sample-level prediction, and $y_s$ represents the sample-level sarcastic label.

During training, a semantic memory implying the inherent understanding for sarcasm expressions will be dynamically maintained to guide the identification of fine-grained sarcasm targets. Within each training mini-batch, we add sample-level features of sarcastic samples (i.e., $\mathbf{S}_i^*$) into a memory buffer $\mathbf{Z} = [\mathbf{S}_1^*, \mathbf{S}_2^*, \cdots, \mathbf{S}_b^*]$, where $b$ denotes the number of sarcastic samples and the notion $*$ is used to mark sarcastic samples. Afterwards, a mean-pooling operation will be applied to the memory buffer and generate the semantic memory $\mathbf{F} \in \mathbb{R}^{1 \times 2d}$, which can be dynamically updated during training by performing the moving average mechanism:

$$\mathbf{F}_t = (1 - \beta) \cdot \mathbf{F}_{t-1} + \beta \cdot \tilde{\mathbf{F}}_t, \quad (2)$$

where $\beta$ is the hyper-parameter for controlling the update degree, $\tilde{\mathbf{F}}_t$ indicates the semantic memory calculated at the t-th iteration, $\mathbf{F}_{t-1}$ and $\mathbf{F}_t$ respectively indicates the dynamically maintained semantic memory updated after $t$-1 and $t$ iterations.

**Table 2: The hyper-parameter settings applied in multimodal models (i.e., UMT [40], MSTI [36] and CofiPara-MSTI [20]). The SSI indicates the sample-level sarcasm identifier.**

| Setting | SaSTI$_{UMT}$ | SaSTI$_{MSTI}$ | SaSTI$_{CofiPara-MSTI}$ |
|---|---|---|---|
| Batch size | 16 | 16 | 2 |
| Epoch number | 40 | 40 | 10 |
| Loss scale $\alpha$ within SSI | 0.438 | 0.577 | 0.460 |
| Memory update parameter $\beta$ | 0.911 | 0.243 | 0.841 |

**MSTI Enhanced by Overall Sarcastic Semantics.** The modeled semantic memory will be utilized to inform specific sarcasm targets respectively within the text and image modality, enabling the fine-grained sarcasm target identification task performed with the guidance of the overall semantics of sarcasms. To this end, $\mathbf{F}_t$ will be respectively projected into the textual and visual space:

$$\mathbf{F}_t^h = \text{Tanh}(\mathbf{F}_t W_1 + b_1),$$
$$\mathbf{F}_t^o = \text{Tanh}(\mathbf{F}_t W_2 + b_2), \quad (3)$$

where $W_1, W_2 \in \mathbb{R}^{2d \times d}$ represent weight parameters, $b_1, b_2 \in \mathbb{R}^{1 \times d}$ represent bias parameters, $\mathbf{F}_t^h \in \mathbb{R}^{1 \times d}$ and $\mathbf{F}_t^o \in \mathbb{R}^{1 \times d}$ represent transformations of the semantic memory $\mathbf{F}_t$ for corresponding modalities. Afterwards, we utilize $\mathbf{F}_t^h$ and $\mathbf{F}_t^o$ to respectively inform textual tokens and visual tokens with close semantic distances towards them. The informed tokens will be then utilized to enhance fine-grained textual and visual features as follows:

$$\mathbf{H}_f = \mathbf{H} + \text{Sim}(\mathbf{H}, \mathbf{F}_t^h) \cdot \text{MLP}(\mathbf{H}),$$
$$\mathbf{O}_f = \mathbf{O} + \text{Sim}(\mathbf{O}, \mathbf{F}_t^o) \cdot \text{MLP}(\mathbf{O}), \quad (4)$$

where Sim represents the cosine similarity function. $\mathbf{H}_f$ and $\mathbf{O}_f$ will be used to implement the final sarcasm target identification.

**Training.** The training objective is mainly two-fold: 1) two major objectives $\mathcal{L}_T$ and $\mathcal{L}_I$ focusing on fined-grained sarcasm target identification respectively for the text and image modality; 2) one auxiliary objective $\mathcal{L}_s$ focusing on sample-level sarcasm identification. Specifically, for the textual sarcasm target identification subtask, the Condition Random Field (CRF) loss between ground-truth labels $Y_{i,j}^T$ and predicted labels $\hat{Y}_{i,j}^T$ will be utilized: $\mathcal{L}_T = \text{CRF}(Y_{i,j}^T, \hat{Y}_{i,j}^T)$, where $\hat{Y}_{i,j}^T$ is generated by applying a linear prediction layer on top of textual features $H_f$. For the visual sarcasm target identification subtask, the Cross-Entropy loss between ground-truth labels $Y_{i,j}^I$ and predicted labels $\hat{Y}_{i,j}^I$ will be utilized: $\mathcal{L}_I = \text{CE}(Y_{i,j}^I, \hat{Y}_{i,j}^I)$, where $\hat{Y}_{i,j}^I$ is generated by applying a linear prediction layer on top of visual features $O_f$. Finally, the auxiliary objective $\mathcal{L}_s$ is implemented as shown in Eq. 1. To sum up, the overall training objective is $\mathcal{L} = \mathcal{L}_t + \mathcal{L}_I + \alpha \cdot \mathcal{L}_s$, where $\alpha$ represents the trade-off hyper-parameter for controlling the contribution of the auxiliary loss.

## 5 EXPERIMENTS

In order to validate the main contributions of this work, we conduct comprehensive performance comparison on both the current MSTI benchmark and the reconstructed MSTI-Plus benchmark.

**Table 3: Performance comparison of the UMT model and the MSTI model trained on different datasets. The results marked with ∗, †, and ‡ are obtained by training in the MSTI benchmark, the sarcasm-only subset of MSTI-Plus benchmark, and the MSTI-Plus benchmark, respectively. All models are tested in the MSTI-Plus benchmark.**

| Model | Textual Sarcasm Target Identification Task | | | Visual Sarcasm Target Identification Task | | |
|---|---|---|---|---|---|---|
| | Micro-F1(%) | Macro-F1(%) | Weighted-F1(%) | Micro-F1(%) | Macro-F1(%) | Weighted-F1(%) |
| MSTI∗ [36] | 26.27 | 29.81 | 29.81 | 13.57 | 11.95 | 3.24 |
| MSTI† [36] | 35.38 | 34.95 | 37.56 | 84.26 | 72.99 | 85.70 |
| MSTI‡ [36] | **60.33** | **54.92** | **60.77** | **89.08** | **76.95** | **89.13** |
| UMT∗ [40] | 26.58 | 30.09 | 30.09 | 13.57 | 11.95 | 3.24 |
| UMT† [40] | 35.02 | 35.52 | 37.16 | 86.67 | 72.75 | 86.94 |
| UMT‡ [40] | **60.66** | **55.81** | **61.35** | **89.39** | **75.53** | **88.95** |

**Table 4: Performance comparison of different approaches based on the MSTI-Plus benchmark.**

| Modality | Model | Multimodal Sarcasm Target Identification | | | | | |
|---|---|---|---|---|---|---|---|
| | | Textual Sarcasm Target Identification Task | | | Visual Sarcasm Target Identification Task | | |
| | | Micro-F1(%) | Macro-F1(%) | Weighted-F1(%) | Micro-F1(%) | Macro-F1(%) | Weighted-F1(%) |
| Text | BiLSTM [11] | 25.45 | 22.64 | 26.47 | - | - | - |
| | BERT [5] | 59.18 | 54.59 | 59.90 | - | - | - |
| Image | ViT [6] | - | - | - | 86.61 | 74.92 | 87.40 |
| | ResNet [12] | - | - | - | 86.73 | 76.23 | 87.74 |
| Multimodal | TPM-MI [13] | 60.76 | 56.03 | 61.46 | 87.15 | 73.65 | 87.39 |
| | MMIB [3] | 60.98 | 55.28 | 61.32 | 89.08 | 77.44 | 89.25 |
| | MSTI [36] | 60.33 | 54.92 | 60.77 | 89.08 | 76.95 | 89.13 |
| | SaSTI$_{MSTI}$ (ours) | **63.46** | **59.14** | **64.28** | **90.53** | **79.07** | **90.35** |
| | UMT [40] | 60.66 | 55.81 | 61.35 | 89.39 | 75.53 | 88.95 |
| | SaSTI$_{UMT}$ (ours) | **61.72** | **57.06** | **62.42** | **90.11** | **78.74** | **90.06** |
| | CofiPara-MSTI [20] | 63.64 | 59.46 | 63.90 | 91.50 | 83.20 | 91.80 |
| | SaSTI$_{CofiPara-MSTI}$ (ours) | **64.55** | **60.35** | **64.48** | **91.80** | **83.55** | **92.04** |

## 5.1 Implementation Details

To extract textual features, we adopt the pre-trained BERT-base-uncased model [5] to process texts. For visual modality, the pre-trained ViT-base model [6] is used to process images. The hyper-parameters used in models are shown in Table 2. The learning rate of the models is set to 5e-5. We use the AdamW optimizer to train the model. The models are trained on a 3090 GPU. In the experiments, we use the Micro-F1, Macro-F1 and Weighted-F1 as the evaluation metrics for the textual sarcasm target identification and the visual sarcasm target identification.

## 5.2 Baselines

In this paper, we compare our approach with text-modality methods, image-modality methods and multimodal methods, which are detailed as follows.

**Text-Modality Methods.** These models identify sarcasm targets by leveraging the sarcastic information from the text modality. We compare with existing text-modality methods, including BiLSTM [11] and BERT [5].

**Image-Modality Methods.** These models focus on mining the sarcasm intention based on the image content to identify whether each visual target is sarcastic. We adopt ResNet [12] and ViT [6] as image-modality baselines.

**Multimodal Methods.** These models mine sarcasm intention by leveraging the semantic information of multimodal samples to identify sarcasm targets and non-sarcasm aspects within texts and images. Our approach compares with existing multimodal sarcasm target identification baselines, including MSTI [36] and CofiPara-MSTI [20]. Moreover, in order to implement the visual sarcasm

target identification, we use the classification head to replace the architecture of object detection within MSTI and CofiPara-MSTI. Besides, due to the similarity between the textual sarcasm target identification task and the named entity recognition task, we also add named entity recognition models (including Unified Multimodal Transformer (UMT) [40], Temporal Prompt Model with Multiple Images (TPM-MI) [13], and MultiModal representation learning with Information Bottleneck (MMIB) [3]) as multimodal baselines.

## 5.3 Results

In this work, we design two sets of experiments to answer two research questions, through which we progressively study the value of MSTI-Plus benchmark and the effectiveness of SaSTI approach:

- **RQ1**: Can non-sarcasm references enhance deep models' ability to identify sarcasm targets and non-sarcasm aspects?

- **RQ2**: Does the proposed approach achieve the superior performance compared to existing baselines?

Next, we detail the answer to each question and discuss experimental results.

**Answer to RQ1**. For RQ1, we conduct experiments on the MSTI-Plus benchmark and the MSTI benchmark. For in-depth analysis, we obtain a subset of MSTI-Plus by removing all non-sarcasm samples in the training set. The subset involves sarcasm samples with fine-grained annotations of sarcasm targets and non-sarcasm aspects. In order to examine whether the non-sarcasm reference enhances the deep models' ability to identify sarcasm targets and non-sarcasm aspects, we train Unified Multimodal Transformer (UMT) [40] and Multimodal Sarcasm Target Identification (MSTI) [36] on three datasets (i.e., the MSTI-Plus, the sarcasm-only subset of MSTI-Plus,

and the MSTI) and then test the UMT model and the MSTI model on the MSTI-Plus. Table 3 shows the performance of UMT model and MSTI model trained on different datasets.

In general, we can draw following observations from Table 3. First, the UMT mdoel and the MSTI mdoel trained on the MSTI benchmark show the poor performances. These results demonstrate that existing models trained on the MSTI benchmark cannot correctly identify sarcasm targets and non-sarcasm aspects. Second, when including the fine-grained supervision of sarcasm targets and non-sarcasm aspects, the UMT model and the MSTI model both obtain clearly improved performances. These results demonstrate that deep models trained on the sarcasm-only subset of MSTI-Plus can perceive the difference between sarcasm targets and non-sarcasm aspects, preventing from incorrectly treating the sarcasm target identification as a common aspect term extraction task. Finally, the UMT model and the MSTI model trained on the MSTI-Plus benchmark can achieve better performances than those trained on the MSTI benchmark and the sarcasm-only subset of MSTI-Plus on all the metrics. These results demonstrate that deep models can better understand the inherent semantics of sarcasms by modeling overall semantics of sarcasm intentions. The above observations clearly show that **the MSTI-Plus benchmark enhances deep models' ability to identify sarcasm targets and non-sarcasm aspects by introducing the non-sarcasm reference.**

**Answer to RQ2**. For RQ2, we compare our SaSTI approach with different baselines, including BiLSTM [11], BERT [5], ResNet [12], ViT [6], TPM-MI [13], MMIB [3], UMT [40], MSTI [36] and CofiPara-MSTI [20]. The corresponding results are shown in Table 4. In general, the following observations are made. First, multimodal methods generally perform better than unimodal methods. The observation demonstrates the necessity of studying multimodal sarcasm target identification. Second, SaSTI attached on top of different multimodal sarcasm target identification models (i.e., MSTI and CofiPara-MSTI) can outperform all baselines. The observation indicates that **the SaSTI approach can inform the overall understanding for sarcastic expressions and make the fine-grained sarcasm target identification well performed with the guidance of the overall understanding for sarcasm intentions.**

## 5.4 Analysis

**Ablation Study.** To further verify the effectiveness of each module within the SaSTI mechanism, we conduct the ablation study for our approach on the MSTI-Plus benchmark and report results in Table 5. The first row of Textual Sarcasm Target Identification (TSTI) task and Visual Sarcasm Target Identification (VSTI) task show the performance of the full model. In the second row of TSTI task and VSTI task, we remove the sample-level sarcasm identifier (SSI) module. We can observe the performance clearly drops, which demonstrates the necessity of using sample-level supervision as guidance to inform the overall understanding for sarcastic expressions. The observation also demonstrates that only using fine-grained supervision signals cannot effectively guide deep models to thoroughly understand the sarcasm semantics, which in turn restricts deep models' ability in the fine-grained sarcasm target identification task. For the last row of the TSTI task and the VSTI

**Table 5: Ablation study results on our constructed benchmark for SaSTI mechanism. The notation "SSI" and "SM" denote sample-level sarcasm identifier and semantic memory.**

| Textual Sarcasm Target Identification Task | | | |
|---|---|---|---|
| | Micro-F1(%) | Macro-F1(%) | Weighted-F1(%) |
| SaSTI (full model) | **62.69** | **59.03** | **63.64** |
| w/o SSI | 60.26 | 56.49 | 61.40 |
| w/o SM | 59.13 | 53.66 | 59.87 |
| Visual Sarcasm Target Identification Task | | | |
| | Micro-F1(%) | Macro-F1(%) | Weighted-F1(%) |
| SaSTI (full model) | **90.71** | **79.74** | **90.60** |
| w/o SSI | 89.63 | 77.55 | 89.55 |
| w/o SM | 89.87 | 78.80 | 89.96 |

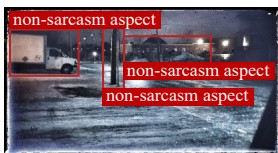 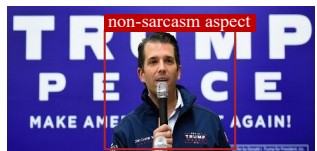

Road looks great. #holdmybeer

[Donald Trump Jr]. Trolls Democrats After They Lose In [Georgia]

**Figure 6: The prediction results of examples. The model trained on the MSTI benchmark identify non-sarcasm aspects (shown in red color) as sarcasm targets.**

task, we remove the semantic memory used to enhance textual features and visual features. The performance degradation observed in the last row clearly validates the effectiveness of semantic memory. By removing the semantic memory, deep models cannot well model the overall semantics of sarcasm intentions and thus show the poor performance for implementing the fine-grained sarcasm target identification.

**Sample Cases.** As shown in Figure 6, there lists prediction results of the model based on the MSTI benchmark. As mentioned in the previous paragraph, the model trained on the MSTI benchmark treats the sarcasm target identification as a common aspect term extraction task and tends to incorrectly recognize non-sarcasm aspects as sarcasm targets.

## 6 CONCLUSION

In this work, we are the first to observe the limitation only containing the fine-grained supervision of sarcasm targets within texts or images in the current MSTI benchmark. Hence, we proposed a more comprehensive benchmark dubbed MSTI-Plus. The main characteristic of MSTI-Plus is to include fine-grained annotations of non-sarcasm aspects into the benchmark. Moreover, we introduce non-sarcasm samples into the MSTI-Plus, aiming to enable the deep model to perceive clear semantics of sarcastic expression. To this end, we proposed a pluggable Semantics-aware Sarcasm Target Identification (SaSTI) mechanism which can be flexibly attached on top of existing multimodal sarcasm target identification models, which can guide the model to clearly perceive the semantic difference between sarcasm targets and non-sarcasm aspects. Extensive experiments demonstrate the effectiveness of the proposed benchmark and SaSTI for identifying sarcasm targets and non-sarcasm aspects.

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

# A  QUALITY CONTROL

In order to minimize the annotation bias due to the subjectivity of annotators, every annotator needs to participate in annotation meetings to discuss how to label sarcasm targets and non-sarcasm aspects within texts and images. Furthermore, to make annotators clearly understand the annotation principle, we allocate each annotator 100 pieces of data consisting of both sarcastic samples and non-sarcastic samples for annotation. Then, we discuss the reason of annotation bias and rectify annotators' misunderstanding for the definition of sarcasm targets and normal aspects. In the annotation

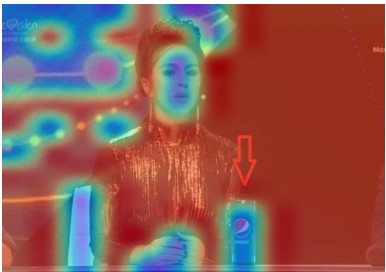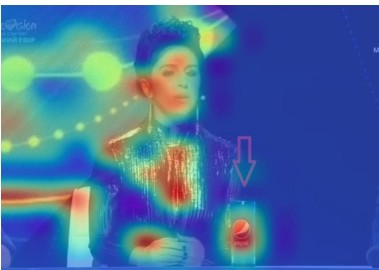

thank god for no product placement in # ukraine # eurovision

Raw Image                                   MSTI                                   SaSTI

**Figure 7: Attention visualization comparison for the MSTI and our approach. The red region represents where the model focuses.**

**Table 6: Performance of the SaSTI mechanism on the twitter-15/17 benchmark. The notation "SSI" and "SM" denote sample-level sarcasm identifier and semantic memory.**

| Aspect-based Sentiment Analysis Task | | |
| --- | --- | --- |
| *Micro-F1(%)* | *Macro-F1(%)* | *Weighted-F1(%)* |
| SaSTI (full model) **56.49** | **53.94** | **56.61** |
| w/o SSI 55.79 | 52.93 | 56.06 |
| w/o SM 54.64 | 51.56 | 54.92 |

process, each sample is labelled by three annotators. If the annotation for one certain sample shows a bias, it will be allocated to other three persons for a second-round annotation agreement process. If the annotation of the re-labeled sample still exists a bias, it will be removed. Finally, we calculate the Cohen's Kappa [25] to measure annotation congruity across annotators. For our annotation process, the kappa score results in 0.806, indicating that our constructed dataset is featured by high-quality annotations.

## B SCALABILITY WITH THE TASK RELATED TO THE ASPECT-BASED SENTIMENT ANALYSIS.

In order to validate the scalability of our proposed SaSTI on other task (i.e., the aspect-based sentiment analysis task), we conduct experiments on the Twitter-15/17 benchmark [22, 44] focusing on identifying the sentiment polarity of the textual aspect. Samples within this dataset have the sample-level sentiment polarity and the fine-grained sentiment polarity for textual tokens. The labels of sentiment polarity have three categories (i.e., positive, negative and neutral). We report results in Table 6. It can be see that our approach demonstrates excellent performance and generalization on other benchmark.

## C ATTENTION VISUALIZATION COMPARISON

Figure 7 displays attention visualizations for our approach and the MSTI by observing the crossmodal interaction between texts and images. The red region represents where the deep model focuses. We can observe that the deep model armed with the SaSTI can effectively perceive the visual region (i.e., "canned kola") which contraries to the textual content (i.e., "no product placement") and the woman that does not convey sarcasm information. However, the

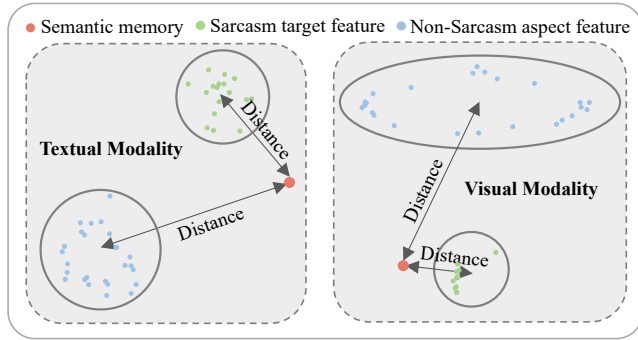

**Figure 8: The t-SNE visualization for the semantic memory, textual aspect features and visual target features.**

baseline badly focuses on the visual region "the woman", rather than the visual region "canned kola" conveying the sarcastic intention. The observation demonstrates our proposed SaSTI helps the model clearly perceives the semantic difference between sarcasm targets and non-sarcasm aspects, which can accurately identify sarcastic and non-sarcastic visual regions.

## D THE T-SNE VISUALIZATION.

In order to measure whether the semantic memory inform specific sarcasm targets of textual tokens or visual object, we show the t-SNE visualization for the semantic memory, textual aspect features and visual target features in Figure 8. We can observe that the distance between the semantic memory and visual or textual sarcasm targets features is close, while the distance to non-sarcasm target features is far. The observation demonstrates the semantic memory can be well inform specific sarcasm targets of textual tokens or visual objects.

