# OpenReview forum: "MSTI-Plus: Introducing Non-Sarcasm Reference Materials to Enhance Multimodal Sarcasm Target Identification"
_ACM.org/TheWebConf/2025/Conference — WWW 2025 Oral_

### Official Review · Reviewer_ViRo · 2024-11-04

**Novelty:** 4
**Technical Quality:** 3

**Review:**

This paper analyzed the shortcomings of the current Multimodal Sarcasm Target Identification (MSTI) benchmark and proposed a new MSTI plus benchmark. Additionally, the authors propose a pluggable Semantics-aware Sarcasm Target Identification (SaSTI) mechanism, which enhances MSTI by incorporating non-sarcastic reference materials, thereby improving the identification of sarcastic elements in multimodal contexts.

Strengths:
1. This paper is meticulously structured with a logical flow that guides the reader through its content. The presentation of figures and tables is appropriate, contributing to the clarity and comprehension of the research findings.
2. This paper analyzed the problem by dissecting the limitations of the existing MSTI benchmark. Not only build a more comprehensive MSTI benchmark but also propose the innovative inclusion of non-sarcastic reference materials to bolster the effectiveness of MSTI.
3. This paper substantiated the arguments with exemplary cases and analyses and analyzed the modules that improve model performance through an ablation study.

Weaknesses:
1. Despite measures to mitigate the subjectivity of annotators, the dataset, which relies on human annotation, may still be prone to biases stemming from varying interpretations among annotators. This potential for annotation bias could influence the model training and the accuracy of the results.
2. While the paper enhances MSTI by introducing non-sarcastic reference materials, the quantity and scope of the dataset might still be limited. The richness and intricacy of sarcastic expressions could potentially exceed the dataset's coverage, constraining the model's ability to generalize to a broader spectrum of sarcastic expressions.
3. The authors have conducted an empirical study; however, a more profound theoretical analysis would strengthen the paper. Such analysis would provide a robust foundation for the rationale behind the proposed method, offering insights into why the SaSTI mechanism is effective and under what conditions it performs optimally.
4. Lacking efficiency analysis of MSTI-Plus and SaSTI: What would be the performance while implementing MSTI-Plus and SaSTI? How much GPU time would this method save, and what kind of results would be achieved compared with the traditional methods mentioned above?
5. The experimental comparisons rely on a closed dataset (MSTI-Plus) without a thorough analysis of its quality, raising concerns about the reliability of the results.

**Questions:**

1. When using the proposed non-sarcasm references, What factors contribute to the performance degradation in sarcasm target identification?
2. Can the SaSTI mechanism be simplified to reduce computational cost and training difficulty while maintaining performance?

**Reviewer Confidence:**

4: The reviewer is certain that the evaluation is correct and very familiar with the relevant literature

**Scope:**

4: The work is relevant to the Web and to the track, and is of broad interest to the community

---

### Official Review · Reviewer_Ae8f · 2024-11-14

**Novelty:** 4
**Technical Quality:** 5

**Review:**

The authors propose a new benchmark by introducing non-sarcastic aspects and non-sarcastic samples as reference information to aid sarcasm recognition and have constructed a new dataset. However, the experiments lack rigorous and transparent comparisons to validate the proposed MSTI-PLUS dataset's advantage over the original MSTI dataset. Specifically, the two datasets differ in the number of sarcastic samples, and training on dataset A while testing on dataset B does not conclusively demonstrate the comparative merits of either dataset.
Additionally, the presentation of experimental results is often unclear. For example, the results for the “full model ” in the ablation study（Table 5) do not align with those in the main experiments(Table 3~4). The hyperparameter settings are numerous, but there is no supplementary information on how they were selected in relation to the experiments.
Lastly, in the area of visual object detection, the authors rely solely on F1 scores and do not account for other previously used metrics such as EM and AP, which would provide a more robust evaluation.

**Questions:**

Lines 582-583: The expression is unclear. Here, $\alpha$ represents the number of image tokens, which should be the number of patches segmented by the ViT model rather than a general indication of quantity.

Lines 789-591, Methodology Section: The explanation of the "Multimodal Transformer Network" lacks clarity. For instance, the authors mention implementing it with the existing MSTI framework, yet it is unclear which specific components were utilized—was it limited to the "MCE" module within MSTI, or did it involve other parts? Although this section may not be central to the paper, the authors should strive for precision in their descriptions.

Experimental Section
Evaluation Metrics: The experiment only reports F1 scores, whereas prior baselines[1,2] included additional metrics such as EM, AP, AP_50, and AP_75. A broader set of evaluation metrics would provide a more comprehensive performance assessment.

Table 3: This table compares results obtained by training on different datasets, raising several concerns:

Fairness of Experimental Comparisons: The comparison is potentially biased, as the datasets differ significantly in sample size and distribution. It is not surprising that a model trained on the MSTI dataset performs lower on the MSTI-PLUS test set than a model trained and tested within the same dataset. The observed performance discrepancies could simply be attributed to the differences in dataset composition rather than model effectiveness.

Differences in MSTI-PLUS (Sarcasm-Only) vs. Full MSTI-PLUS: According to Table 1, the training sample counts differ markedly between these two settings, making it difficult to discern whether the performance gains are due to the model observing more samples or benefiting from the inclusion of non-sarcastic examples.

Table 5 (Full Model of SaSTI): Clarification is needed on the definition of the "full model" mentioned here. The results for this "full model" do not appear in the primary experiment tables (Table 3 and Table 4), leading to confusion about its composition and relevance.


[1]Multimodal Sarcasm Target Identification in Tweets
[2]CofiPara: A Coarse-to-fine Paradigm for Multimodal Sarcasm Target Identification with Large Multimodal Models

**Reviewer Confidence:**

3: The reviewer is confident but not certain that the evaluation is correct

**Scope:**

3: The work is somewhat relevant to the Web and to the track, and is of narrow interest to a sub-community

---

### Official Review · Reviewer_ap97 · 2024-11-29

**Novelty:** 6
**Technical Quality:** 6

**Review:**

Summary
This paper extends multimodal sarcasm target identification (MSTI; [1]) by:
- Reformulating the task into target classification (as justified in Section 3) rather than bounding box detection.
- Introducing non-sarcasm target labels for both textual and visual modalities.
- Creating a fine-grained, manually annotated dataset, MSTI-Plus.
The authors also propose a novel, pluggable approach, SaSTI, to enhance existing methods by incorporating auxiliary sample-level features via a dynamically maintained semantic memory.

Strengths
- The paper identifies significant gaps in the original MSTI framework and provides well-justified alternative approaches.
- The inclusion of fine-grained annotations for both sarcastic and non-sarcastic elements improves interpretability and might also address overgeneralization.
- The authors provide detailed descriptions of their methodologies and conduct extensive experiments under various settings, ensuring clarity and reproducibility.
- The release of code and, notably, the MSTI-Plus dataset is a valuable contribution that can serve as a strong benchmark for sarcasm target identification tasks.

Weaknesses
- The distinction between B-/I-Normal and O labels is somewhat unclear. For instance, in Figure 2, it is ambiguous why "an hour" is labeled as Inside while "$15" is labeled as Outside. This lack of clarity could potentially lower agreement between annotators.
- The approach requires additional computational resources for training, extracting, and providing auxiliary sample-level features from semantic memory.
- While the method introduces a novel perspective, there are abundant pre-existing sample-level datasets (e.g., listed in [2]) that might be used to augment the proposed semantic memory mechanism approach.

**Questions:**

- Resource Usage and Complexity: An analysis of the computational complexity and resource usage of the proposed SaSTI approach would be appreciated. This could include runtime comparisons with baseline models, GPU memory requirements, and the scalability of semantic memory for larger datasets.
- Task Reformulation and Overlap: By reconfiguring the task from bounding box detection to target classification, the nature of the task may have fundamentally changed. It would strengthen the paper to quantify the overlap between the original MSTI and MSTI-Plus datasets, particularly in the image modality, to ensure the task's reformulation still captures the same conceptual goals.
- Leveraging Existing Multimodal Datasets: Several multimodal datasets already exist to inform binary sample-level sarcasm identification [2]. It might be beneficial to explore whether integrating these datasets at scale could help compress additional auxiliary information into the SaSTI framework and enhance its effectiveness.
- Exploring Subjectivity and Disagreements: While the study aims to provide a high-quality sarcasm target identification dataset with a high level of intercoder agreement, it would be insightful to analyze the nature of disagreements from the annotation process. The two-step annotation agreement procedure rejected 15% of the dataset, suggesting that these disagreements might provide valuable insights into the subjective nature of sarcasm detection.
- VinVL Boundary Conditions: Are there any specific boundary conditions where VinVL fails to detect sarcasm targets but should have? For example, are there sarcasm targets that are inherently ambiguous or highly contextual in visual modality? Could these cases inform future improvements in visual sarcasm target identification?

[1] Wang, J., Sun, L., Liu, Y., Shao, M., & Zheng, Z. (2022, May). Multimodal sarcasm target identification in tweets. In Proceedings of the 60th Annual Meeting of the Association for Computational Linguistics (Volume 1: Long Papers) (pp. 8164-8175).
[2] Farabi, S., Ranasinghe, T., Kanojia, D., Kong, Y., & Zampieri, M. (2024). A Survey of Multimodal Sarcasm Detection. arXiv preprint arXiv:2410.18882.

**Reviewer Confidence:**

3: The reviewer is confident but not certain that the evaluation is correct

**Scope:**

3: The work is somewhat relevant to the Web and to the track, and is of narrow interest to a sub-community

---

### Official Review · Reviewer_NMPV · 2024-11-30

**Novelty:** 3
**Technical Quality:** 3

**Review:**

**Pros**
1. Non-sarcasm reference is important for multimodal sarcasm target identification.
2. A benchmark is proposed in this paper, with both sarcasm targets and normal non-sarcasm aspects.
__________

**Cons**
1. The positioning of this paper is unclear. If it is a research paper, it should focus on describing the model part (Section 4.4) and discard Section 3 or put it in the experiment part. The model proposed in this paper is more like a simple adjustment of existing methods on this dataset.
2. However, this paper seems to focus on the construction and advantages of the benchmark, it does not seem suitable for this track.
3. In addition, there is a lack of multi-dimensional evaluations of the quality of the benchmark.

**Questions:**

Please refer to the cons.

**Reviewer Confidence:**

2: The reviewer is willing to defend the evaluation, but it is likely that the reviewer did not understand parts of the paper

**Scope:**

3: The work is somewhat relevant to the Web and to the track, and is of narrow interest to a sub-community